# Interaction-induced nematic Dirac semimetal from quadratic band touching: A constrained-path quantum Monte Carlo study

Zi Hong Liu,[1] Hongyu Lu,[2] Zi Yang Meng,[2] and Lukas Janssen[1]

[1]*Institut für Theoretische Physik and Würzburg-Dresden Cluster of Excellence ct.qmat, TU Dresden, 01062 Dresden, Germany*
[2]*Department of Physics and HKU-UCAS Joint Institute of Theoretical and Computational Physics,*
*The University of Hong Kong, Pokfulam Road, Hong Kong, China*
(Dated: September 11, 2025)

Electronic systems with quadratic band touchings, commonly found in two- and three-dimensional materials such as Bernal-stacked bilayer graphene, kagome metals, HgTe, and pyrochlore iridates, have attracted significant interest concerning the role of interactions in shaping their electronic properties. However, even in the simplest model of spinless fermions on a two-dimensional checkerboard lattice, the quantum phase diagram as a function of nearest-neighbor interaction remains under debate. We employ constrained-path quantum Monte Carlo simulations (CP-QMC) simulations to investigate the problem using a two-dimensional torus geometry. We cross-validate our results on small lattices by comparing them with density-matrix renormalization group calculations, finding quantitative agreement. In particular, we implement an improved optimization scheme within the CP-QMC simulations, enabling the identification of a bond-nematic Dirac semimetal phase that was found in tensor-network studies on cylindrical geometries, but remains inaccessible to Hartree-Fock mean-field methods. The CP-QMC approach makes it possible to establish the emergence of this phase in a geometry that preserves lattice rotational symmetry and permits extrapolation to the thermodynamic limit. Our results show that the quantum phase diagram of spinless fermions on the checkerboard lattice with nearest-neighbor repulsion features three interaction-induced phases at half filling: a quantum anomalous Hall insulator at weak coupling, a bond-nematic Dirac semimetal at intermediate coupling, and a site-nematic insulator at strong coupling.

## I. INTRODUCTION

Interacting electron systems with isolated Fermi points represent a central platform for investigating emergent quantum many-body phenomena [1, 2]. The quantum phase diagrams of such systems host a variety of nontrivial electronic phases, including interaction-induced topological phases such as topological insulators [3, 4] and topological semimetals [5, 6], as well as non-Fermi liquids [7], phases with unconventional symmetry breaking [8–10], and semimetallic phases exhibiting emergent symmetries [11, 12]. Furthermore, a variety of exotic quantum phase transitions have been identified in these systems, including semimetal-to-insulator transitions exhibiting emergent relativistic invariance [13, 14] or even supersymmetry [15–17], as well as unconventional order-to-order transitions that lie beyond the Landau paradigm [18–22]. While many of these developments were initially motivated by theoretical interest and explored in simplified toy models, recent work has proposed microscopic realizations via twist-tuning in moiré materials, including twisted bilayer graphene [23, 24] and twisted double-bilayer transition metal dichalcogenides [25].

On the theoretical front, a major challenge is that many models exhibiting rich quantum many-body phenomena are hindered by the sign problem in quantum Monte Carlo simulations [26]. As a result, one often has to rely on biased methods, uncontrolled approximations, or numerical simulations restricted to very small lattices. In such cases, confidence in theoretical predictions can only be achieved by converging results from competing, complementary approaches in an integrated manner.

In the present work, we seek to realize such convergence for the case of spinless fermions on the checkerboard lattice with nearest-neighbor repulsion [27]. This model represents an ideal test bed for evaluating the validity of various quantum many-body methods, as its proposed ground-state phase diagram is sufficiently complex to make comparisons nontrivial, yet simple enough to allow convergence.

The checkerboard lattice consists of two sublattices 1 and 2, depicted as blue and yellows balls in Fig. 1(a). In the non-interacting limit, the model features a single quadratic band touching point in the electronic spectrum, located at the corner $(\pi, \pi)$ of the first Brillouin zone, see Fig. 1(b). At half filling, the Fermi level sits right at the touching point. Importantly, on the checkerboard lattice, the quadratic band touching is symmetry protected, namely by time reversal and a four-fold lattice rotational symmetry with rotation center located at the midpoint between two sites of the same sublattice, as indicated by the gray dotted lines in Fig. 1(a). Note that this situation is different from the quadratic band touching point occurring in the nearest-neighbor hopping model on the Bernal-stacked honeycomb bilayer, which allows for the emergence of interaction-induced Dirac cones without spontaneous symmetry breaking [11, 12]. Upon the inclusion of interactions, however, the ground state may break some of these symmetries spontaneously, allowing the emergence of a finite gap. In fact, previous renormalization group studies [8, 27, 28] showed that the leading instability for weak repulsive interactions corresponds to a quantum anomalous Hall (QAH) phase, characterized by a full gap in the fermionic spectrum, but with gapless edge modes arising as a consequence of the nontrivial topology of the electronic bands. The QAH state features spontaneously generated currents, as illustrated in Fig. 1(c). In this phase, time reversal symmetry is spontaneously broken, but lattice symmetries remain preserved. At strong coupling, by contrast, the ground state breaks the four-fold lattice rotational symmetry, but preserves time reversal symmetry, realizing a site-nematic insulator (SNI) [29]. In this state, the charge den-

sity on one sublattice is different from the charge density on the other sublattice, as depicted in Fig. 1(e).

The phase diagram at intermediate coupling, however, has remained a matter of some debate. While initial mean-field results [27] suggested an intermediate mixed phase in which both time reversal and lattice rotational symmetry are spontaneously broken, exact diagonalization studies suggested a direct first-order transition from the QAH phase at small coupling and the SNI phase at large coupling [30]. Initial density matrix renormalization group (DMRG) calculations [29, 31] were consistent with the direct-transition scenario; however, other DMRG studies [32] suggested a novel intermediate bond-nematic Dirac semimetal (BNDS) phase, which respects time reversal and as such is distinct from the mixed phase obtained in mean-field theory [27]. The BNDS phase is illustrated in Fig. 1(d). A more recent study that employed, besides DMRG, also exponential and tangent space renormalization group calculations [33] revealed that the BNDS phase at intermediate coupling breaks the *same* four-fold lattice rotational symmetry as the site-nematic insulator at strong coupling, rendering the corresponding semimetal-to-insulator transition a quantum analog of the classical liquid-gas transition.

In DMRG and related tensor-network methods, extrapolation to the thermodynamic limit is typically performed using cylindrical lattice geometries, which often exhibit significant anisotropy in the length-to-width ratio. While such a setup may be adequate for studying the competition between states that preserve lattice rotational symmetry in the thermodynamic limit, greater care is required when nematic states are among the possible ground states. This is because significant length-to-width anisotropy in the cylindrical lattice setup can bias the finite-size system toward nematic order. In this work, we aim to uncover the phase diagram using a square lattice geometry that preserves lattice rotational symmetry even at finite sizes. At intermediate and strong coupling, the model exhibits a severe negative-sign problem in standard determinant quantum Monte Carlo (DQMC) simulations [26]. To address this challenge, we employ the constrained-path quantum Monte Carlo (CP-QMC) method in combination with the branching random walker algorithm [34, 35]. The CP-QMC method has previously been successfully applied to Hubbard-type models on the square lattice [36–38] and cross-validated with DMRG and DQMC (in the parameters where the sign-problem is not severe) on various geometries, i.e. serving as the successful example of the integrated approach for challenging 2d quantum many-body systems.

In CP-QMC, the ground-state projection is carried out in Slater determinant space, where an ensemble of walkers evolves under imaginary-time propagation. This formulation avoids the sign problem encountered in DQMC simulations, at the cost of introducing a bias that depends on the quality of the trial wavefunction. The trial state serves as the starting point for an iterative self-consistency optimization procedure [39, 40]. In this work, we use DMRG calculations on small lattices to assess the quality of our trial wavefunction. We find that quantitative agreement with DMRG results is achieved when self-consistent CP-QMC simulations are initialized from multiple starting states drawn from a class of site-

independent mean-field solutions. This agreement on small lattices gives us confidence that the CP-QMC results remain reliable on larger systems. We perform simulations on square lattices with up to $N = 2 \times 16^2$ sites, significantly surpassing the system sizes accessible to previous tensor-network-based methods. We find that the BNDS phase is stabilized over an extended range of couplings between the QAH phase at weak coupling and the SNI phase at strong coupling. Notably, this range broadens as the lattice size increases, supporting the phase's persistence in the thermodynamic limit, we summarize the obtained phase diagram in Fig. 1(f). Our results offer the integrated solution of the phase diagram of spinless fermions on 2d checkerboard lattice, as a function of nearest-neighbor interaction.

The remainder of this paper is organized as follows. In Sec. II, we introduce the model. Section III details the numerical techniques employed in this study: we first analyze the sign structure within DQMC simulations and then present our CP-QMC approach, to overcome the sign problem. Our results, based on site-independent mean-field theory as a starting point for the self-consistency optimization procedure in CP-QMC and small-size DMRG simulations are presented in Sec. IV. We discuss our findings and outline future directions in Sec. V.

## II. MODEL

We consider the following microscopic Hamiltonian defined on a two-dimensional checkerboard lattice,

$$H = H_0 + H_V \tag{1}$$

with

$$H_0 = -t_1 \sum_{\mathbf{r},\delta} \left( c_{\mathbf{r},1}^\dagger c_{\mathbf{r}+\delta,2} + \text{h.c.} \right)$$
$$- t_2 \sum_{\mathbf{r},\lambda,i} (-1)^{\lambda+i} \left( c_{\mathbf{r},\lambda}^\dagger c_{\mathbf{r}+\mathbf{a}_i,\lambda} + \text{h.c.} \right) \tag{2}$$

$$H_V = V \sum_{\mathbf{r},\delta} \left( c_{\mathbf{r},1}^\dagger c_{\mathbf{r},1} - \frac{1}{2} \right) \left( c_{\mathbf{r}+\delta,2}^\dagger c_{\mathbf{r}+\delta,2} - \frac{1}{2} \right) \tag{3}$$

where $\lambda = 1, 2$ labels the two sublattices within each unit cell at position $\mathbf{r}$. The vector $\delta$ connects a site on sublattice 1 to its nearest neighbors on sublattice 2. The primitive lattice vectors are given by $\mathbf{a}_1 = (1, 0)$ and $\mathbf{a}_2 = (0, 1)$. Throughout this work, we set the nearest-neighbor hopping amplitude $t_1 = 1$ and the sublattice- and direction-dependent next-nearest-neighbor hopping $t_2 = 0.5$. The interaction strength $V$ is treated as a tuning parameter that drives the quantum phase transitions of the model. The model is illustrated in Fig. 1(a).

In momentum space, the fermion bilinear part of the Hamiltonian, $H_0$, features a gapless quadratic band touching point located at the edge $\mathbf{k} = (\pi, \pi)$ of the first Brillouin zone, see Fig. 1(b). Unlike Dirac points, which feature linear dispersion, the band touching point displays a quadratic dispersion $\sim \mathbf{q}^2$, where $\mathbf{q}$ corresponds to the distance from $\mathbf{k} = (\pi, \pi)$ in momentum space, leading to a finite density of states at the Fermi

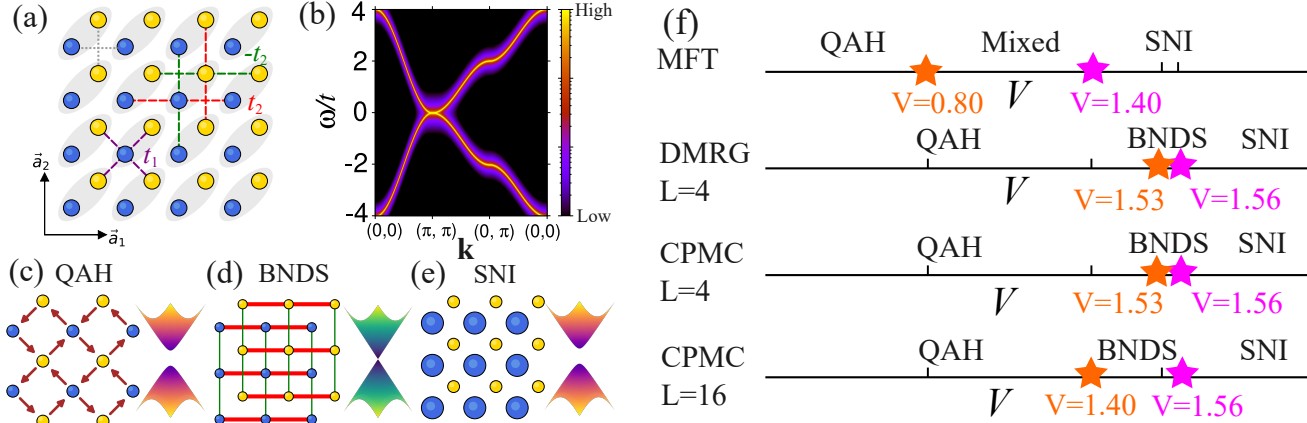

FIG. 1. (a) Illustration of the $t_1$-$t_2$-$V$ model of spinless fermions on the checkerboard lattice. Purple, red, and green dashed lines correspond to nearest- and next-nearest-neighbor hoppings $t_1$, $t_2$, and $-t_2$, respectively. Gray dashed lines indicate the $C_4$ lattice rotational symmetry of the model. (b) Single-particle spectral function along a high-symmetry path in the first Brillouin zone in the noninteracting limit $V = 0$. The quadratic band touching point is located at $\mathbf{k} = (\pi, \pi)$. (c) Real-space symmetry-breaking pattern of the QAH phase at small coupling. Arrows indicate spontaneously generated currents. The electronic spectrum, schematically shown in the inset, is characterized by a full gap as a consequence of the spontaneously broken time reversal symmetry. (d) Same as (c), but for the BNDS phase at intermediate coupling. Thick red (thin green) lines indicate enhanced (suppressed) hoppings on next-nearest-neighbor bonds. The electronic spectrum is characterized by a two Dirac cones located along the edge of the first Brillouin zone, as a consequence of the spontaneously broken $C_4$ lattice rotational symmetry. (e) Same as (c), but for the SNI phase at strong coupling. Large blue (small yellow) balls indicate enhanced (suppressed) charge densities on the two different sublattices. The electronic spectrum is characterized by a full gap, arising from an annihilation of the two Dirac cones in the BNDS phase. The SNI phase breaks the same $C_4$ lattice rotational symmetry as the BNDS phase. (f) Zero-temperature phase diagram of the model as a function of nearest-neighbor repulsion $V$, from site-independent mean-field theory (MFT), density matrix renormalization group (DMRG) calculations on a square lattice geometry with linear system size $L = 4$, and constrained-path quantum Monte Carlo (CP-QMC) simulations on a square lattice geometry with linear system sizes $L = 4$ and $L = 16$. We use periodic boundary conditions in each case. Mean-field theory predicts an intermediate mixed phase in which both time reversal and $C_4$ lattice rotational symmetry is spontaneously broken. The DMRG simulations on small lattices instead find an intermediate BNDS phase. The CP-QMC simulations quantitatively agree with the DMRG simulations on small lattices. For larger lattice sizes, the BNDS phase is stabilized over an extended range of couplings, supporting its persistence in the thermodynamic limit.

level. This enhanced degeneracy renders the system highly susceptible to interaction-driven instabilities.

## III. METHODS

In our integrated appraoch, we combine large-scale quantum Monte Carlo simulations with density matrix renormalization group (DMRG) calculations to study this system. While previous works have provided DMRG results on cylinder geometries, in this study we additionally perform DMRG simulations on small torus clusters as benchmarks. To access larger system sizes on torus geometries, we focus on quantum Monte Carlo simulations. Although the standard DQMC method based on the Blankenbecler-Scalapino-Sugar algorithm provides an unbiased approach [41, 42], it suffers from a severe fermion sign problem [26] at strong coupling (large $V$), making it impractical in this regime. To overcome this challenge, we adopt the constrained-path approximation combined with the branching random walker algorithm [34, 35]. This approach allows us to effectively suppress the sign problem and obtain reliable results deep in the interacting regime. For comparison, we also implement a site-independent mean-field approach, which not only aids in optimizing the CP-QMC simulations but also provides a connection to previous studies on this problem [27].

Details of the numerical methods are provided in the following subsections.

### A. Site-independent mean-field theory

Performing a Hartree-Fock-type mean-field analysis prior to large-scale lattice simulations is often valuable, as it provides intuitive insight into the possible phase diagram of the system. In this work, we also use the mean-field result as a starting point to initialize the self-consistency procedure in the CP-QMC simulations.

We employ a site-independent mean-field analysis for the model defined in Eq. (1). Within the mean-field approximation, we can write the Hamiltonian as

$$H_{\text{MFT}} = H_0 + V \sum_{\mathbf{r},\delta} \left[ n_{\mathbf{r},1} \langle n_{\mathbf{r}+\delta,2} \rangle + n_{\mathbf{r}+\delta,2} \langle n_{\mathbf{r},1} \rangle \right. $$
$$\left. - \frac{1}{2} \left( n_{\mathbf{r},1} + n_{\mathbf{r}+\delta,2} \right) - \langle n_{\mathbf{r}+\delta,2} \rangle \langle n_{\mathbf{r},1} \rangle + \frac{V}{4} \right], \quad (4)$$

where $n_{\mathbf{r},\lambda} = c_{\mathbf{r},\lambda}^\dagger c_{\mathbf{r},\lambda}$ is the local occupation operator and $\langle n_{\mathbf{r},\lambda} \rangle$ is the corresponding local density order parameter. The numerical minimization of the mean-field Hamiltonian for a

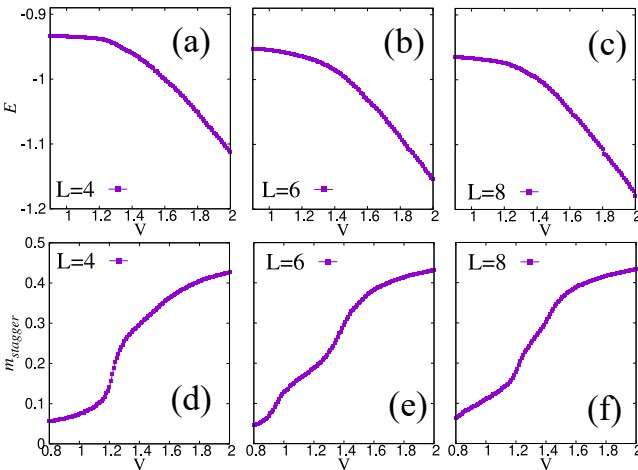

FIG. 2. (a) Single-site energy $E$ as a function of $V$ from site-independent mean-field theory with open boundary conditions along the $y$ direction and periodic boundary conditions for the other direction, for linear system size $L = 4$. (b) Same as (a), but for $L = 6$. (c) Same as (a), but for $L = 8$. (d), (e) and (f) are the same as (a), (b) and (c), but for the sstaggered CDW order parameter $m_{\text{stagger}}$ as functions of the interaction strength $V$, for system size $L = 4, 6, 8$.

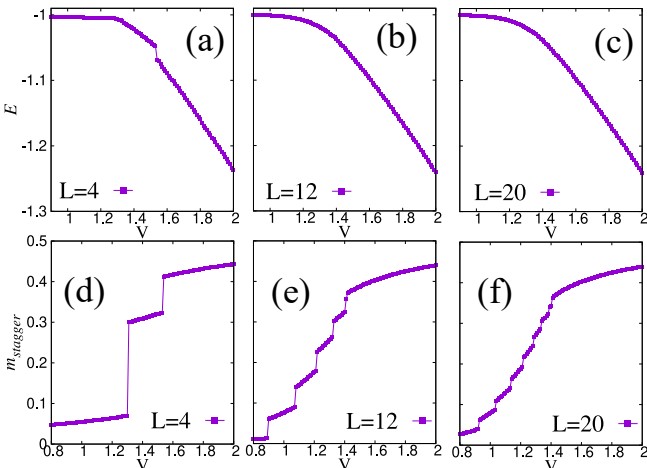

FIG. 3. Same as Fig. 2, but using periodic boundary conditions and system sizes (a,d) $L = 4$, (b,e) $L = 12$, and (c,f) $L = 20$.

given value of $V$ in Eq. (4) provides the corresponding mean-field ground state, characterized by the real-space distribution of $\langle n_{\mathbf{r},\lambda} \rangle$.

As an illustrative example we show in Figs. 2(a–c) the single-site energy as a function of $V$ for different system sizes $L$, within mean-field theory with open boundary conditions along the $y$ direction. Figures 2(d–f) show the corresponding staggered density defined as

$$ m_{\text{stagger}} = \frac{1}{N_{\text{s}}} \sum_{\mathbf{r},\lambda} (-1)^{\lambda} n_{\mathbf{r},\lambda}, \qquad (5) $$

which serves as an order parameter for the site-nematic insulator phase. Here, $N_{\text{s}}$ corresponds to the number of sites. Importantly, in the finite-size system with open boundary conditions, the staggered density varies smoothly with $V$. The smooth variation of the staggered magnetization indicates a continuous connection between the noninteracting quadratic band touching state at $V = 0$ and the strongly-interacting state at $V \to \infty$ for the finite-size system. This continuity ensures that the iterative CP-QMC procedure (for a given value of $V$), initialized with the mean-field solution as the trial wavefunction, converges to a unique final state, regardless of which mean-field solution (obtained from the same or other values of $V$) is used.

This is different from the situation using periodic boundary conditions, shown in Fig. 3. For instance, for $L = 4$, the staggered density exhibits two discrete step-like jumps, see Fig. 3(d). These jumps correspond to the cusp and discontinuity in the energy curve shown in Fig. 3(a). As the system size increases to $L = 12$ and $20$ [Figs. 3(b) and (c)], the energy curves become smoothly decreasing functions of $V$. Meanwhile, the staggered density order parameter [Figs. 3(e) and (f)] shows an increasing number of step-like jumps with system

size, while the magnitude of each individual jump diminishes. In contrast to the smooth behavior under open boundary conditions, the step-like jumps cause the final state obtained in CP-QMC to depend strongly on the choice of mean-field trial wavefunction used for initialization. This also implies that the CP-QMC approach in the case of periodic boundary conditions may struggle to reach the true ground state when the initial trial wavefunction is far from optimal, often requiring additional iterations or multiple runs. To address this convergence challenge, we initiate *multiple* CP-QMC self-consistency cycles for each parameter set $(L, V)$, using trial states generated from mean-field solutions at various values of $V$. These include at least one representative from each distinct regime separated by the jumps shown in Figs. 3(d–f). Upon convergence, the run that produces the lowest ground-state energy is selected as the definitive result for that $(L, V)$ parameter set.

### B. Sign problem in determinant quantum Monte Carlo

For zero-temperature simulations, we may employ standard DQMC method based on the Blankenbecler-Scalapino-Sugar algorithm [43, 44] to obtain the ground state via imaginary-time projection, $|\psi_0\rangle = \lim_{\Theta \to \infty} e^{-\Theta H} |\psi_I\rangle$, where $|\psi_I\rangle$ is a trial wavefunction that has non-zero overlap with the true ground state $|\psi_0\rangle$. The projection operator is discretized as $e^{-\Theta H} = \prod_{n=1}^{M} e^{-\Delta\tau H} \approx \prod_{n=1}^{M} e^{-\Delta\tau H_V} e^{-\Delta\tau H_0}$, where $\Delta\tau = \Theta/M$ is the Trotter time step. To treat the four-fermion interaction terms, we apply a Hubbard-Stratonovich transformation. The interaction can be rewritten as

$$ V \left( c_{\mathbf{r},1}^{\dagger} c_{\mathbf{r},1} - \frac{1}{2} \right) \left( c_{\mathbf{r}+\delta,2}^{\dagger} c_{\mathbf{r}+\delta,2} - \frac{1}{2} \right) - \frac{V}{4} = -\frac{V}{2} O_{\mathbf{r},\delta}^2 \qquad (6) $$

where the operator $O_{\mathbf{r},\delta}$ takes different forms depending on the Hubbard-Stratonovich channel, $O_{r,\delta} = n_{\mathbf{r},1} - n_{\mathbf{r}+\delta,2}$ for the $S_z$ (charge) channel and $O_{r,\delta} = c_{\mathbf{r},1}^{\dagger} c_{\mathbf{r}+\delta,2} + \text{h.c.}$ for the $S_x$ channel. This allows the evolution operator to be expressed as a

path integral over auxiliary fields $\{x_{\mathbf{r},\delta}\}$, with each exponential decoupled as $e^{O^2/2} = \frac{1}{\sqrt{2\pi}} \int dx \, e^{-\frac{x^2}{2}+xO}$. The full evolution becomes

$$\prod_{n=1}^{M} e^{-\Delta\tau H} \propto \prod_{n=1}^{M} \int d\mathbf{x}^n \, p\left(\mathbf{x}^n\right) B\left(\mathbf{x}^n\right) \tag{7}$$

where $p(x)$ is the standard Gaussian distribution and $B\left(\mathbf{x}^n\right)$ is the fermionic propagator at time slice $n$ under the field configuration $\mathbf{x}^n$.

The auxiliary fields $\{x_{\mathbf{r},\delta}^n\}$ are sampled using the Metropolis algorithm to compute ensemble averages of observables. For an operator $A$, its expectation value is evaluated as

$$\langle A \rangle = \frac{\int \mathcal{D}\mathbf{x} W\left(\mathbf{x}\right) A\left(\mathbf{x}\right)}{\int \mathcal{D}\mathbf{x} W\left(\mathbf{x}\right)}, \tag{8}$$

with $W\left(\mathbf{x}^i\right) = p\left(\mathbf{x}^i\right) \langle\psi_I| \prod_{i=1}^{M} B\left(\mathbf{x}^i\right) |\psi_0\rangle$, where $\mathcal{D}\mathbf{x}$ represents integration over all auxiliary field configurations. In practice, the fermionic part of the weight, $\langle\psi_I| \prod_{i=1}^{M} B\left(\mathbf{x}^i\right) |\psi_0\rangle$, can be negative or complex, leading to the fermion sign problem. In such cases, one must perform reweighting, and the severity of the sign problem is quantified by the average sign,

$$\langle\text{sign}\rangle = \frac{\int \mathcal{D}\mathbf{x}\,\text{sign}\left\{W\left(\mathbf{x}^i\right)\right\} \left|W\left(\mathbf{x}^i\right)\right|}{\int \mathcal{D}\mathbf{x} \left|W\left(\mathbf{x}^i\right)\right|}. \tag{9}$$

In Fig. 4, we show the average sign as a function of interaction strength $V_1$ for system sizes $L = 4, 6, 8$, using both $S_z$ and $S_x$ Hubbard-Stratonovich decomposition channels. While both channels suffer from the sign problem at large $V_1$, the $S_z$ channel generally provides a higher average sign across the same parameter range. In the CP-QMC simulations discussed below, we apply the Hubbard-Stratonovich decomposition in the $S_z$ channel, where the bias introduced by the constraint is expected to be reduced. Notably, for $V_1 \geq 1.4$, the average sign for $L \geq 6$ drops below 0.1, making unbiased Blankenbecler-Scalapino-Sugar simulations numerically unstable and unreliable for large systems. Interestingly, as we will show later, this interaction strength roughly coincides with the onset of a phase transition into the bond-nematic Dirac semimetal (BNDS) phase.

### C. Constrained-path quantum Monte Carlo

To address the sign problem, we implement the constrained-path quantum Monte Carlo (CP-QMC) method [34], formulated as a constrained approximation within the branching random walker algorithm. This approach reformulates the ground-state projection in Slater determinant space, where an ensemble of walkers evolves under imaginary-time propagation. Within this scheme, observables are evaluated as weighted averages over the walker ensemble. For instance, the mixed estimator of an operator $A$ is given by

$$\langle A \rangle_{\text{mix}} = \frac{\langle\psi_T| A |\psi_0\rangle}{\langle\psi_T|\psi_0\rangle} = \frac{\langle\psi_T| A e^{-\Theta\hat{H}} |\psi_I\rangle}{\langle\psi_T| e^{-\Theta\hat{H}} |\psi_I\rangle}, \tag{10}$$

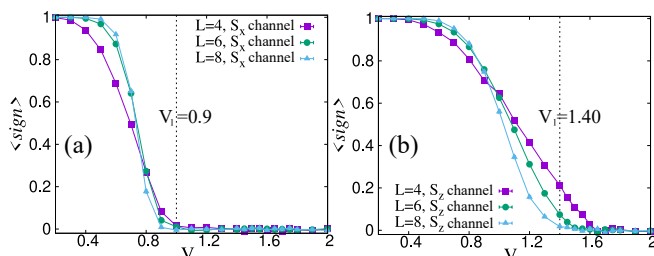

FIG. 4. (a) Average sign as a function of $V$ for system size $L = 4, 6, 8$ for Hubbard-Stratonovich decoupling in the $S_x$ channel. The projection length is fixed at $\Theta = 6$. The vertical black dotted line marks the largest $V$ for which the Blankenbecler-Scalapino-Sugar algorithm yields a tolerable average sign, $\langle\text{sign}\rangle \geq 0.1$, for $L \geq 6$. (b) Same as (a), but for Hubbard-Stratonovich decoupling in the $S_z$ (charge) channel, yielding a less severe sign problem.

where $\psi_T$ is the trial wavefunction. The above mixed estimator is used to evaluate observables that commute with the Hamiltonian, such as the total energy and total density. For general operators that do not commute with the Hamiltonian, we employ the backpropagation technique to obtain real estimators with improved accuracy. In this approach, the bra state of the ground-state expectation value is constructed by propagating backward in imaginary time, using a fixed auxiliary field configuration sampled from the forward propagation path,

$$\langle A \rangle_{\text{real}} = \frac{\langle\psi_T| e^{-\tau_{\text{BP}}\hat{H}} A |\psi_0\rangle}{\langle\psi_T| e^{-\tau_{\text{BP}}\hat{H}} |\psi_0\rangle}. \tag{11}$$

where $\tau_{\text{BP}}$ denotes the backpropagation time. We apply this real estimator to measure the order parameters. In CP-QMC simulations, we set $\tau_{\text{BP}} = 2$, which turns out to be sufficient to obtain converged results.

The initial wave function may be expanded in a basis of Slater determinants as

$$|\psi_I\rangle = \sum_k \omega_k^0 \left|\phi_k^0\right\rangle, \tag{12}$$

where $\left|\phi_k^0\right\rangle$ is a slater determinant, and $\omega_k^0$ is the corresponding weight. $k$ labels the different walkers. For each walker $k$, the projected overlap can be expressed as

$$\langle\psi_T|e^{-\Theta\hat{H}} |\psi_I\rangle$$
$$\propto \omega_k^0 \int \mathcal{D}\mathbf{x} \prod_{n=1}^{M} p\left(\mathbf{x}^n\right) \langle\psi_T| \prod_{n=1}^{M} B\left(\mathbf{x}^n\right) \left|\phi_k^0\right\rangle$$
$$= \langle\psi_T|\phi_k^0\rangle \sum_k \omega_k^0 \int \mathcal{D}\mathbf{x} \prod_{n=1}^{M} C_n \left\{ \frac{p\left(\mathbf{x}^n\right)}{C_n} \frac{\langle\psi_T|\phi_k^n\rangle}{\langle\psi_T|\phi_k^{n-1}\rangle} \right\}$$
$$= \langle\psi_T|\phi_k^0\rangle \sum_k \omega_k^n \prod_{n=1}^{M} \int d\mathbf{x}^n \Omega\left(\mathbf{x}^n|\mathbf{x}^{n-1}\right) \tag{13}$$

where we have used the Hubbard-Stratonovich decomposition introduced in Eq. (7). The propagated Slater determinant at

time slice $n$ is defined as $\left|\phi_k^n\right\rangle = \prod_{i=1}^n B\left(\mathbf{x}^i\right)\left|\phi_k^0\right\rangle$, and the updated weight is $\omega_k^n = \omega_k^0 \prod_{n=1}^M C_n$, with $C_n$ being a normalization factor arising from importance sampling. Equation (13) defines the conditional probability distribution used in the branching random walker algorithm. At each time slice $n$, the ratio of overlaps $\frac{\langle\psi_T|\phi_k^n\rangle}{\langle\psi_T|\phi_k^{n-1}\rangle}$ modifies the standard Gaussian distribution of auxiliary fields into a conditioned distribution $\Omega\left(\mathbf{x}^n\middle|\mathbf{x}^{n-1}\right)$, normalized by $C_n$. The normalization factor is absorbed into the walker's weight $\omega_k^n$, reflecting the role of importance sampling in guiding the stochastic evolution. The cumulative configuration $\left\{\mathbf{x}^1, \mathbf{x}^2, \ldots, \mathbf{x}^n\right\}$ is sampled from the sequence of conditional distributions $\Omega\left(\mathbf{x}^n\middle|\mathbf{x}^{n-1}\right)$, and represents the path of a random walker in the configuration space.

In the presence of a sign problem, the overlap ratio $\frac{\langle\psi_T|\phi_k^n\rangle}{\langle\psi_T|\phi_k^{n-1}\rangle}$ is no longer a positive real number, rendering the conditional probability distribution ill-defined. To resolve this, a constraint $\langle\psi_T|\phi_k^n\rangle = 0$ is imposed to terminate any path whose overlap with the trial wavefunction becomes non-positive. This approximation eliminates the sign problem and becomes exact in the limit where the trial wavefunction coincides with the true ground state [34], which is generally unknown. Therefore, the accuracy of the simulation depends on the quality of the chosen trial wavefunction, which controls the bias introduced by the constraint.

To balance accuracy and computational efficiency, we use single Slater determinants as trial wavefunctions, serving as the starting point for an iterative procedure in the CP-QMC simulations. The iterative procedure allows us to optimize the trial wavefunction for each set of physical parameters until self-consistency has been reached [39, 40]. Figure 5 shows a flowchart illustrating the self-consistency convergence process. Starting from an initial guess, such as a free-fermion Slater determinant or a mean-field solution, we compute the mixed estimator of the single-particle Green's function $G_{ij}^{\mathrm{mix}} = \left\langle c_i^\dagger c_j\right\rangle_{\mathrm{mix}}$ at each iteration step using the trial wavefunction from the previous step. The updated trial wavefunction is then constructed from the eigenvector matrix $U$ obtained by diagonalizing the mixed Green's function, $G^{\mathrm{mix}} = U\lambda U^\dagger$. After a few iterations, this self-consistent procedure yields an optimized trial wavefunction that is adapted to the physical parameters and improves the accuracy of the simulation.

In our study of the model defined in Eq. (1), we found that the self-consistent approach starting from a single initial trial wavefunction systematically improves the accuracy of CP-QMC results in quasi-one-dimensional simulations. However, in two-dimensional torus geometries, the self-consistency procedure can suffer from convergence issues, often getting trapped in local minima due to degeneracies arising from translational symmetry. To overcome this issue, we perform multiple self-consistent simulations starting from different initial states, selected from a class of site-independent mean-field solutions. The final result is chosen as the converged solution with the lowest total energy among all runs. This is illustrated in Figs. 6(a–c), which show the evolution of energy as a function of iteration steps using CP-QMC simulations for a fixed value of $V$, initialized with *multiple* mean-field solutions

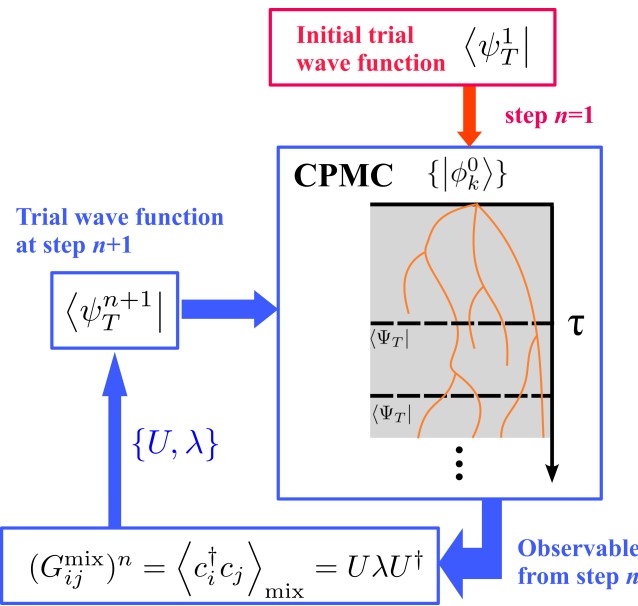

FIG. 5. Flowchart illustrating the self-consistency convergence process. In the first iteration $n = 1$, an initial trial wavefunction $\Psi_T^1$ is provided to the CP-QMC simulation. At each step $n$, the mixed estimator of the single-particle Green's function $G_{\mathrm{mix}}^n$ is measured, and the updated trial wavefunction for iteration $n+1$ is obtained using the eigenvectors $U$ and eigenvalues $\lambda$ of $G_{\mathrm{mix}}^n$.

obtained from different values of $V$. The runs do not necessarily converge to the same energy values. In some cases they coincide, but more often they differ, underscoring the need to compare all outcomes. As shown in Fig. 6(d), the energy curves corresponding to different initial trials intersect. To identify the correct ground-state solution at a given value of $V$ in the CP-QMC simulation, we compare the final energies from all runs and select the lowest-energy result as the best estimate for the true ground state.

## IV. RESULTS

In general, the CP-QMC simulations starting from multiple single-Slater-determinant trial wavefunctions provide us with an estimate for a member of the ground-state manifold. In the case of a symmetry-breaking ground state, the long-range order can be directly identified through the measurement of corresponding order parameters. As discussed earlier, we are particularly interested in the quantum anomalous Hall (QAH) phase, the bond-nematic Dirac semimetal (BNDS), and the site-nematic insulator (SNI), as well as in whether a mixed phase, as predicted by mean-field theory, can be stabilized.

### A. Order parameters

The QAH phase breaks time-reversal symmetry, and its corresponding order parameter is defined through the current-current correlation function,

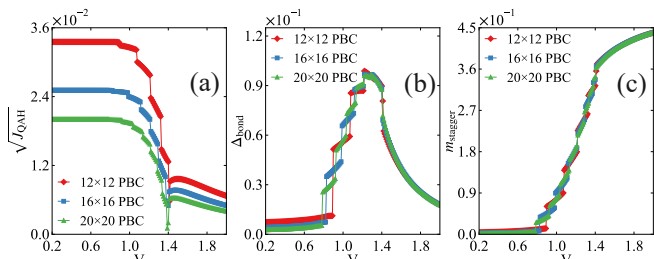

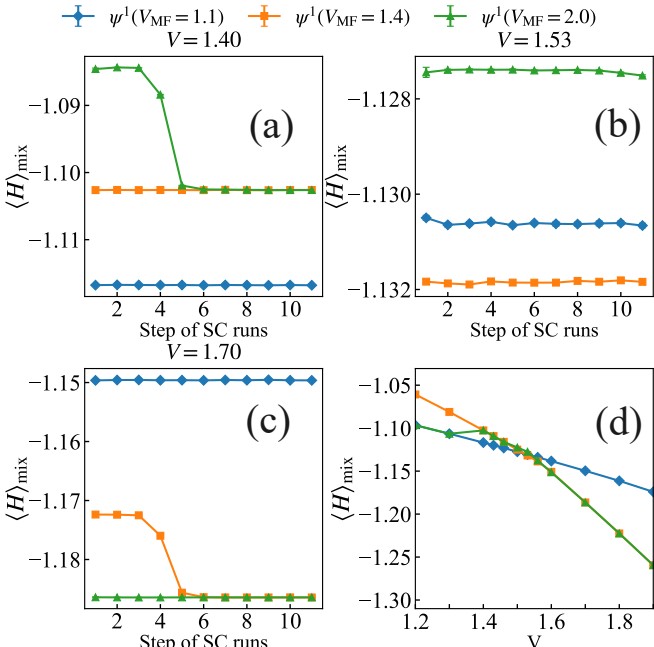

FIG. 7. (a) QAH order parameter $\sqrt{\mathcal{J}_{\text{QAH}}}$ as a function of $V$ for lattice sizes $L$, from site-independent mean-field theory with periodic boundary conditions. (b) Same as (a), but for the bond nematic order parameter $\Delta_{\text{bond}}$. (c) Same as (a), but for the staggered density order parameter $m_{\text{stagger}}$.

FIG. 6. (a) Evolution of the energy expectation value as a function of iteration steps using CP-QMC simulations for fixed $V = 1.4$, initialized with *multiple* mean-field solutions obtained from different values of $V$. The simulations are done for $L = 4$ with periodic boundary conditions in both directions. Blue, orange, and green curves correspond to initial trial states drawn from mean-field theory at $V_{\text{MF}} = 1.1, 1.4,$ and 2.0. (b) Same as (a), but for $V = 1.53$. (c) Same as (a), but for $V = 1.70$. (d) Converged energy expectation value obtained after 11 self-consistency steps as a function of $V$. The lowest-energy state obtained from the multiple CP-QMC simulations is taken as the best estimate for the true ground state.

between the two sublattices,

$$\delta_{\text{SNI}} = \frac{1}{N_s} \sum_{\mathbf{r}} \left\langle \left( c_{\mathbf{r},1}^{\dagger} c_{\mathbf{r},1} - c_{\mathbf{r},2}^{\dagger} c_{\mathbf{r},2} \right) \left( c_{r_0,1}^{\dagger} c_{r_0,1} - c_{r_0,2}^{\dagger} c_{r_0,2} \right) \right\rangle, \tag{16}$$

where $\mathbf{r}$ runs over all unit cells, $r_0$ denotes a reference unit cell, $N_s$ is the total number of sampling sites. This order parameter characterizes the imbalance of electron density between the two sublattices (labeled '1' and '2'), and a nonzero value signals an imbalance between the two sublattices, which spontaneously breaks the lattice rotational symmetry.

Since the BNDS and SNI phases break the same symmetries, the bond nematic order parameter $\Delta_{\text{bond}}$ will be finite in the SNI phase as well, and the site nematic order parameter $\delta_{\text{SNI}}$ will be finite also in the BNDS phase. However, as we show below, the transition from BNDS to SNI has a clear first-order nature, across which $\sqrt{\mathcal{J}_{\text{QAH}}}$ jumps from large to small values, while $\delta_{\text{SNI}}$ jumps from small to large values.

$$\mathcal{J}_{\text{QAH}} = \frac{1}{N_b} \sum_{\mathbf{r}, \boldsymbol{\delta}} \epsilon_{\mathbf{r}, \boldsymbol{\delta}} \langle \mathcal{C}_{\mathbf{r}, \boldsymbol{\delta}} \mathcal{C}_{\mathbf{r}_0, \boldsymbol{\delta}_0} \rangle \tag{14}$$

where $\mathcal{C}_{\mathbf{r}, \boldsymbol{\delta}} = i c_{\mathbf{r},1}^{\dagger} c_{\mathbf{r}+\boldsymbol{\delta},2} + \text{h.c.}$ is the bond current operator, $\epsilon_{\mathbf{r}, \boldsymbol{\delta}}$ characterizes the orientation of the current. $N_b$ denotes the total number of bonds in the system. $\mathbf{r}_0, \boldsymbol{\delta}_0$ indicate the reference bond.

In our analysis, we take the square root $\sqrt{\mathcal{J}_{\text{QAH}}}$ as the QAH order parameter.

The BNDS phase breaks $C_4$ rotational symmetry. To detect this phase, we evaluate the anisotropy in the next-nearest-neighbor hopping amplitudes, characterized by the bond order parameter,

$$\Delta_{\text{bond}} = \frac{1}{2} \sum_{\lambda} \left| \left\langle c_{\mathbf{r}, \lambda}^{\dagger} c_{\mathbf{r}+\mathbf{a}_1, \lambda} \right\rangle \right| - \left| \left\langle c_{\mathbf{r}, \lambda}^{\dagger} c_{\mathbf{r}+\mathbf{a}_2, \lambda} \right\rangle \right|, \tag{15}$$

where $\mathbf{a}_1$ and $\mathbf{a}_2$ are the lattice unit vectors along the $x$ and $y$ directions.

The SNI phase features a sizable density-density correlation

## B. Site-independent mean field solution

Figure 7 shows the QAH order parameter $\sqrt{\mathcal{J}_{\text{QAH}}}$, the bond nematic order parameter $\Delta_{\text{bond}}$ and the staggered density $m_{\text{stagger}}$ as a function of $V$ for different system sizes. For small values of $V \lesssim 0.8$, the nematic order parameter and the staggered density vanishes in the thermodynamic limit, indicating a QAH ground state. At large values of $V \gtrsim 1.4$, the QAH order parameter becomes small, indicating a site-nematic insulator ground state. At intermediate values of $V$ between $0.8 \lesssim V \lesssim 1.4$, however, all three order parameter tend to a finite value in the thermodynamic limit, implying a mixed phase in which both time reversal and lattice rotational symmetry is spontaneously broken. This result is in agreement with the mean-field analysis of Ref. [27]. The corresponding phase diagram is shown in Fig. 1(f).

## C. Quasi-one-dimensional CP-QMC simulation

We first apply CP-QMC to a quasi-one-dimensional cylindrical geometry with open boundary conditions along the $y$

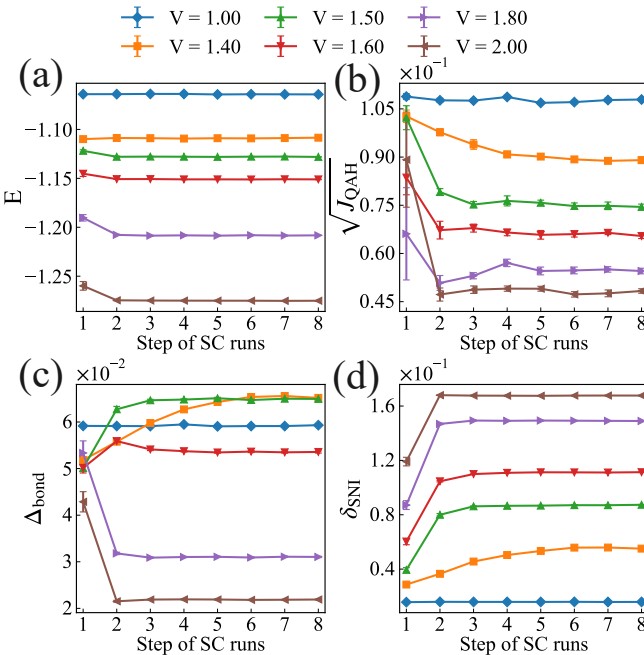

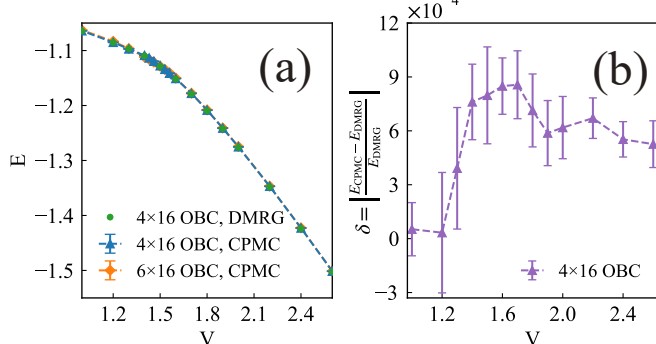

FIG. 9. (a) Single-site energy as a function of interaction strength $V$, , obtained from DMRG and CP-QMC simulations with open boundary conditions (OBC). (b) Relative difference between the DMRG and CP-QMC energies as a function of $V$ on a $4 \times 16$ checkerboard lattice with open boundary conditions.

FIG. 8. (a) Single-site energy, (b) QAH order parameter $\sqrt{\mathcal{J}_{QAH}}$, (c) BNDS order parameter $\Delta_{\text{bond}}$, and (d) SNI order parameter $\delta_{\text{SNI}}$ as a function of steps of self-consistency runs. Different curves correspond to different interaction strength $V$.

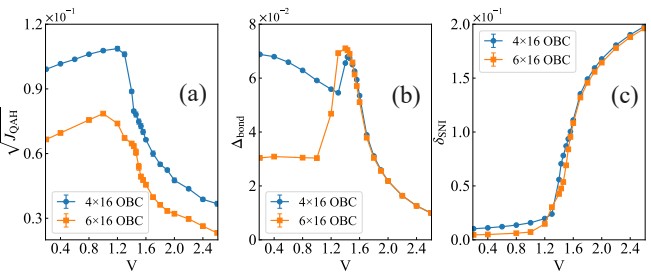

FIG. 10. (a) QAH order parameter $\sqrt{\mathcal{J}_{QAH}}$ (b) BNDS order parameter $\Delta_{\text{bond}}$, and (c) SNI order parameter $\delta_{\text{SNI}}$ as functions of $V$, computed on a quasi-one-dimensional checkerboard lattice with open boundary conditions along the $y-$direction using CP-QMC simulation.

direction and periodic boundary conditions along the $x$ direction. As in standard DMRG practice, we choose $L_y \gg L_x$. Physical observables are measured only in the central region of the cylinder, defined by $\frac{L_y - L_x}{2} < y < \frac{L_y + L_x}{2}$, ensuring that boundary influences are minimized.

In Fig. 8, we show the convergence of physical observables during the CP-QMC self-consistent optimization, using a Slater determinant of the quadratic band touching state as the initial trial wavefunction. As discussed in Sec. III, the self-consistency procedure under open boundary conditions converges to the same solution regardless of the initial trial state. As illustrated in Fig. 8, the ground-state energy converges within 2–3 iterations for all values of $V$, while the order parameters require up to 7–8 iterations to reach full convergence. In Fig. 9, we compare the single-site energy as a function of $V$ obtained from CP-QMC and DMRG on a $4 \times 16$ cylinder. After self-consistency optimization, we achieve a relative energy error below $10^{-3}$ between the two methods.

Figure 10 shows the order parameters from CP-QMC with self-consistency on $4 \times 16$ and $6 \times 16$ cylinders. The results reproduce the phase diagram previously found by DMRG [32, 33]. At weak coupling, a finite QAH order parameter confirms the stability of the QAH phase. Around $V \sim 1.1$, a first-order transition into the BNDS phase is signaled by a discontinuous jump in the BNDS order parameter. At stronger coupling $V > 1.7$, the system enters the SNI phase, with the BNDS order parameter smoothly vanishing as the SNI order parameter grows. While we do not extract precise phase boundaries here, these benchmarks demonstrate excellent agreement between CP-QMC and DMRG on cylinder

geometries.

### D. Two-dimensional simulations with periodic boundary condition

We extend the CP-QMC method to the two-dimensional model with square geometry and periodic boundary conditions. Based on our mean-field analysis, we perform multiple self-consistency runs, each initialized from trial states drawn from the different regimes identified in the mean-field solution, to overcome convergence issues in the self-consistency procedure. As an illustrative example, we focus on a CP-QMC simulations on a $4 \times 4$ lattice with PBC.

Following this procedure, we compare observables on the $4 \times 4$ lattice with periodic boundary conditions obtained from CP-QMC and DMRG simulations. As shown in Fig. 11(a–b), the CP-QMC energies agree with DMRG to within a relative error of $10^{-3}$, demonstrating excellent consistency at this system size. Furthermore, the order parameters in Fig. 11(c) reveal two first-order phase transitions: from the QAH phase to the BNDS phase at $V = 1.53$ and from the BNDS phase to the SNI phase at $V = 1.56$. These discontinuous jumps in the QAH, BNDS, and SNI order parameters match the DMRG-

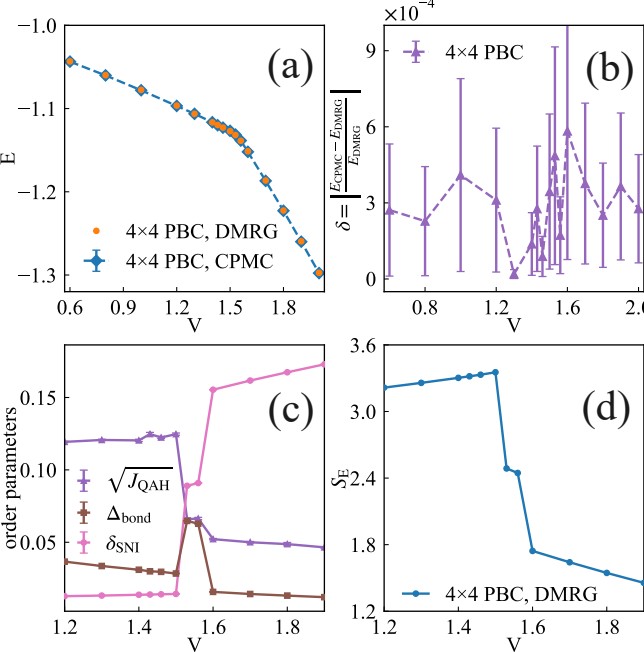

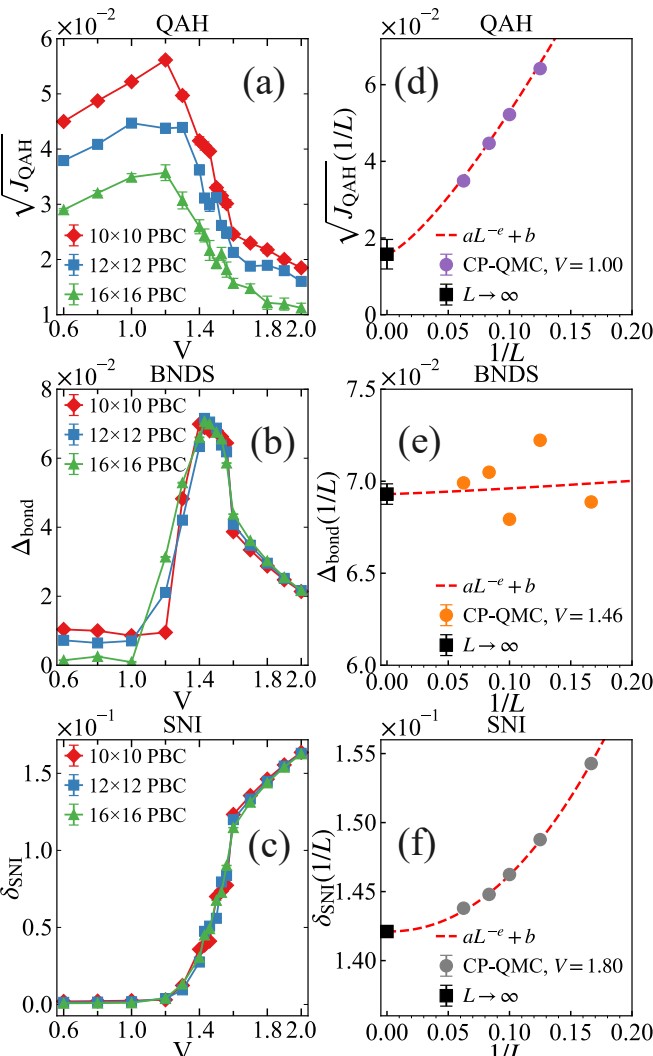

FIG. 11. (a) Single-site energy as a function of interaction strength $V$, obtained from DMRG and CP-QMC simulations with periodic boundary conditions (PBC). (b) Relative difference between the DMRG and CP-QMC energies as a function of $V$. (c) QAH order parameter $\sqrt{\mathcal{J}_{QAH}}$, BNDS order parameter $\Delta_{\mathbf{bond}}$, and SNI order parameter $\delta_{\mathbf{SNI}}$ as functions of $V$ obtained from CP-QMC simulation. (d) Bipartite entanglement entropy $S_E$ as function of $V$ obtained from DMRG. All results are obtained on a $4 \times 4$ checkerboard lattice.

FIG. 12. (a-c) QAH order parameter $\sqrt{\mathcal{J}_{QAH}}$, (b) BNDS order parameter $\Delta_{\mathbf{bond}}$, and (c) SNI order parameter $\delta_{\mathrm{SNI}}$ as functions of $V$, computed on a two-dimensional checkerboard lattice with periodic boundary conditions using CP-QMC simulation. (d-f)Finite-size scaling of the same order parameters versus $1/L$. Red dotted line are power-law fits of the form $aL^{-e} + b$, and black symbols mark the extrapolated thermodynamic-limit values ($1/L = 0$).

determined phase boundaries. Similarly, Fig. 11(d) shows the bipartite entanglement entropy $S_E$ exhibiting step-like jumps at $V = 1.53$ and $V = 1.56$, further confirming the consistency of the transition points in both methods.

Finally, we extend the CP-QMC simulations to larger torus geometries. Figure 12(a-c) shows the evolution of the order parameters with increasing system size: the two first-order transitions remain sharply defined even as $L$ grows. Notably, the BNDS phase boundary shifts toward weaker coupling at larger system size, stabilizing in the window $V = 1.4$ to $V = 1.6$.

To assess the robustness of each phase, we perform finite-size scaling (Fig. 12)(d-f) using the power-law ansatz $O(1/L) = O(1/L = 0) + a(1/L)^e$, which extrapolates each order parameter to the thermodynamic limit. At $V = 1.46$ in the BNDS phase, Fig. 12(e) suggests that the BNDS order parameter becomes essentially size-independent, yielding $\Delta_{\mathbf{bond}}(1/L = 0) = 0.0693(5)$ in the thermodynamic limit, underscoring the robustness of the Dirac semimetal phase. At $V = 1.80$ (SNI phase), Fig. 12(f) confirms a clear power-law scaling, with $\delta_{\mathbf{SNI}}(1/L = 0) = 0.1421(2)$. At $V = 1.00$ (QAH phase), the QAH order parameter, $\sqrt{\mathcal{J}_{QAH}}$, follows a decaying power law but extrapolates to a finite value $\sqrt{\mathcal{J}_{QAH}} = 0.015(3)$, see Fig. 12(d).

It is worth emphasizing that our CP-QMC simulations use only real-valued parameters, both in the Hubbard-Stratonovich

decomposition and in the initial Slater-determinant trial wavefunctions, so no explicit time-reversal-symmetry-breaking terms are present. Nonetheless, spontaneous time-reversal symmetry breaking in the QAH phase is clearly revealed by the bond current-current correlation function $\epsilon_{\mathbf{r},\delta}\langle C_{\mathbf{r},\delta}C_{\mathbf{r}_0,\delta_0}\rangle$, as defined in Eq. 14. To illustrate the nature of the QAH phase, in Fig. 13, we plot these correlations in real space, fixing the reference bond at the lattice center (marked by a purple star). Deep in the QAH phase [$V = 1.00$, Fig. 13(a)], the correlations remain long-ranged, whereas in the SNI phase [$V = 2.00$, Fig. 13(b)], they decay rapidly. Together, these results confirm the robustness of the QAH, BNDS, and SNI phases in the thermodynamic limit.

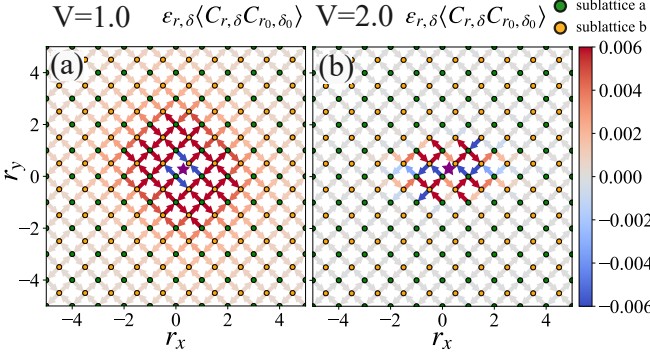

FIG. 13. Bond current-current correlation function $\epsilon_{ij} \left\langle C_{ij} C_{i_0 j_0} \right\rangle$ in real space for $L = 16$, $V = 1.0$ and $V = 2.0$. Arrows indicate the orientation of each bond current, and their color intensity represents the correlation strength with the reference bond (marked by the purple star).

## V.  DISCUSSION

We have extended the numerical study of the model of interacting spinless fermions on the checkerboard lattice to large-scale two-dimensional lattices with square geometry and periodic boundary conditions. To overcome the sign problem in standard determinant quantum Monte Carlo simulations, we have applies the constrained-path quantum Monte Carlo (CP-QMC) method, combined with multiple self-consistency optimizations. To assess the validity of the approach, we have benchmarked our results against DMRG on cylinder geometries and small-size torus geometries, yielding quantitative agreement. Our integrated approach indicate that the phase diagram identified by DMRG remains robust on the full two-dimensional lattice. Notably, we observe that the bond-nematic Dirac semimetal phase broadens toward weaker coupling as system size increases, ultimately stabilizing in a sizable window. This behavior underscores the robustness of the intervening gapless Dirac phase.

Our results show that the CP-QMC approach can be used to quantitatively assess the phase diagram of an interacting fermion model, even in the case when the true ground state of the system is not captured within mean-field theory. In the present case, this happens for the intermediate Dirac semimetal phase, which is not stabilized within previous [27] and our mean-field approach. Nevertheless, initiating the CP-QMC simulations with the incorrect mean-field states still gives the correct ground state consistent with DMRG. For this, however, it has proven important to employ initialize the CP-QMC

simulations for a given fixed parameter set with multiple trial wavefunctions, obtained from mean-field theory from various different parameter sets.

In terms of the experimental realization of quadratic band touching systems, besides the commonly found Bernal-stacked bilayer graphene, kagome metals, HgTe, and pyrochlore iridates, there is also explicit proposal that single layer of $CrCl_2(pyrazine)_2$ might realize the situation of quadratic band touching protected by $C_4$ symmetry [45]. It will be interesting to see whether the interaction-driven electronic states discussed here will be detected in these systems.

In a broader perspective, our results might give some confidence that the method, using CP-QMC simulations initialized with multiple trial wavefunctions, may be successfully applied also in other situations plagued by a sign problem in standard determinant quantum Monte Carlo simulations. In particular, it would be interesting to test whether the method may be applicable to study the physics of the fractional quantum anomalous Hall state found in the model away from integer band filling [46, 47], or to study the interplay between the proposed Bose metal state [48] and competing superconducting states [49] in a Hubbard-type model with spin-dependent anisotropic hoppings.

## ACKNOWLEDGMENTS

We thank Shiwei Zhang and Mingpu Qin for valuable discussions on the technical implementation of the CP-QMC method. This work has been supported by the Deutsche Forschungsgemeinschaft through Project No. 247310070 (SFB 1143, A07), Project No. 390858490 (Würzburg-Dresden Cluster of Excellence *ct.qmat*, EXC 2147), and Project No. 411750675 (Emmy Noether program, JA2306/4-1). The authors gratefully acknowledge the computing time made available to them on the high-performance computer at the NHR Center of TU Dresden. This center is jointly supported by the German Federal Ministry of Education and Research and the state governments participating in the NHR [50]. HYL and ZYM acknowledge the support from the Research Grants Council (RGC) of Hong Kong (Project Nos. AoE/P-701/20, 17309822, HKU C7037-22GF, 17302223, 17301924), the ANR/RGC Joint Research Scheme sponsored by RGC of Hong Kong and French National Research Agency (Project No. A HKU703/22). They also thank HPC2021 system under the Information Technology Services at the University of Hong Kong, as well as the Beijng PARATERA Tech CO.,Ltd. (URL: https://cloud.paratera.com) for providing HPC resources that have contributed to the research results reported in this paper.

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
