# Peer review of "Interaction-induced nematic Dirac semimetal from quadratic band touching: A constrained-path quantum Monte Carlo study"

_SciPost Physics_

## Round 1 · Referee Report · Anonymous (Referee 1) · 2025-10-2

Report

In this paper, the authors investigate a model of spinless fermions on the checkerboard lattice (that is characterized by a quadratic band touching in the noninteracting limit) with nearest-neighbor Coulomb repulsion. The study is primarily based on constrained-path quantum Monte Carlo simulations, with mean-field analysis and DMRG also employed. Their results, benchmarked against DMRG on quasi-1d geometries, show the existence of three different phases as a function of $V$: A quantum anomalous Hall phase, a bond-nematic Dirac semimetallic phase, and a site-nematic insulator. The three phases are distinguished with the help of appropriate correlators, and the corresponding order parameter is extrapolated in the thermodynamic limit.

The main result of the paper is the stabilization, on relatively large two-dimensional clusters, of the bond-nematic Dirac phase, which does not appear already at the mean-field level, but was obtained within DMRG on cylindrical geometries.

Since this paper does not predict a new phase, but rather provides more solid numerical evidence of the bond-nematic phase's existence, the numerical evidence must be robust. In this respect, my primary concern is with the approach used by the authors, namely, performing multiple self-consistency runs initialized from different mean-field trial states, and then selecting the one with the lowest energy as the final result. This approach seems in contradiction with the fact that the constrained-path Monte Carlo (CPMC) mixed estimator for the energy is not an upper bound to the exact energy, that is, CPMC is not a variational approach where the lowest energy corresponds to the best approximation of the true ground state (see for instance Carlos, Gubernatis, Ortis, Zhang, PRB 59, 12788 (1999)). This point requires clarification for the validity of the paper itself.

Once this important point is fixed, there are a few more questions that should be addressed by the authors, as listed below:

-) In the conclusions and also in the introduction, the authors refer to a certain number of experimental realizations of quadratic band touching systems. I would suggest expanding this discussion a bit, discussing whether any material is characterized by a checkerboard lattice, and to what extent the results obtained on the checkerboard lattice can also be applied to other lattice geometries.

-) In the Hartree-Fock decoupling of Eq. 4, there are no terms like $\langle c^{\dagger}_{r,1} c_{r+\delta,2}\rangle c^{\dagger}_{r+\delta,2} c_{r,1}$ and $\langle c^{\dagger}_{r+\delta,2} c_{r,1}\rangle c^{\dagger}_{r,1} c_{r+\delta,2}$ that renormalize the electron hopping and might be relevant in principle. Did the authors try to include them in their mean-field analysis, or do they have some argument about their non-relevance in the mean-field calculations?

-) I think it would be beneficial for the reader to include, in Sec. IIIB, some more details about the Hubbard-Stratonovich decomposition in the $S_z$ channel.

-) At the end of Sec. IVA, the authors state that the transition from the BNDS to the SNI phase shows a jump in $\sqrt{\cal{J}_{\textrm{QAH}}}$, while I think that they refer to a jump in $\Delta_{\textrm{bond}}$.

-) The entanglement entropy $S_E$ is shown in a figure, but it is not defined in the text.

Finally, I would like to mention a couple of typos/inaccuracies that I encountered in the text:

-) There is one additional $V$ inside the square bracket of Eq. 4.

-) Toward the end of page 8, there is written "agree with DMRG to within".

-) In the introduction, where the authors say "Upon the inclusion of interactions, however, the ground state may break some of the symmetries", it is unclear if they refer to the checkerboard lattice or to the Bernal-stacked honeycomb layer.

Recommendation

Ask for major revision

---

## Editorial Decision

awaiting_resubmission